# A Novel Metastatic Estrogen Receptor-Expressing Breast Cancer Model with Antiestrogen Responsiveness

**DOI:** 10.3390/cancers15245773

**Published:** 2023-12-09

**Authors:** Kendall L. Langsten, Lihong Shi, Adam S. Wilson, Salvatore Lumia, Brian Westwood, Alexandra M. Skeen, Maria T. Xie, Victoria E. Surratt, JoLyn Turner, Carl D. Langefeld, Ravi Singh, Katherine L. Cook, Bethany A. Kerr

**Affiliations:** 1Department of Cancer Biology, Wake Forest University School of Medicine, Winston-Salem, NC 27157, USA; klangste@wakehealth.edu (K.L.L.); lihong.shi@nih.gov (L.S.); slumia@wakehealth.edu (S.L.); amskeen@wakehealth.edu (A.M.S.); mtxie@wakehealth.edu (M.T.X.); victoria.e.surratt@vumc.org (V.E.S.); jolturne@wakehealth.edu (J.T.); rasingh@wakehealth.edu (R.S.); klcook@wakehealth.edu (K.L.C.); 2Department of Surgery, Wake Forest University School of Medicine, Winston-Salem, NC 27157, USA; awilson@wakehealth.edu (A.S.W.); bwestwoo@wakehealth.edu (B.W.); 3Department of Biostatistics and Data Science, Wake Forest University School of Medicine, Winston-Salem, NC 27157, USA; clangefe@wakehealth.edu; 4Wake Forest Baptist Comprehensive Cancer Center, Winston-Salem, NC 27157, USA

**Keywords:** estrogen receptor alpha, metastasis, bone metastasis, antiestrogen treatment

## Abstract

**Simple Summary:**

Most women diagnosed with breast cancer (BC) have estrogen receptor-alpha positive (ER+) disease. The current mouse models of ER+ BC often rely on supplemented estrogen to encourage metastasis, which modifies the immune system and the function of some tissues (like bone), or use genetically modified or immunocompromised mouse strains, which do not accurately replicate the clinical disease. We developed a mouse model of triple-negative (TN) breast cancer with virally transduced ER expression that metastasizes spontaneously without exogenous estrogen stimulation and is responsive to antiestrogen drugs. Our mouse model exhibited upregulated ER-responsive genes and multi-organ metastasis without exogenous estrogen administration. Following antiestrogen treatment (tamoxifen, ICI 182,780, or vehicle control), tumor volumes and weights were significantly decreased, exemplifying antiestrogen responsivity. This tumor model, which expresses the estrogen receptor, metastasizes spontaneously, and responds to antiestrogen treatment, will allow for further investigation into the biology and potential treatment of metastasis.

**Abstract:**

Most women diagnosed with breast cancer (BC) have estrogen receptor alpha-positive (ER+) disease. The current mouse models of ER+ BC often rely on exogenous estrogen to encourage metastasis, which modifies the immune system and the function of some tissues like bone. Other studies use genetically modified or immunocompromised mouse strains, which do not accurately replicate the clinical disease. To create a model of antiestrogen responsive BC with spontaneous metastasis, we developed a mouse model of 4T1.2 triple-negative (TN) breast cancer with virally transduced ER expression that metastasizes spontaneously without exogenous estrogen stimulation and is responsive to antiestrogen drugs. Our mouse model exhibited upregulated ER-responsive genes and multi-organ metastasis without exogenous estrogen administration. Additionally, we developed a second TN BC cell line, E0771/bone, to express ER, and while it expressed ER-responsive genes, it lacked spontaneous metastasis to clinically important tissues. Following antiestrogen treatment (tamoxifen, ICI 182,780, or vehicle control), 4T1.2- and E0771/bone-derived tumor volumes and weights were significantly decreased, exemplifying antiestrogen responsivity in both cell lines. This 4T1.2 tumor model, which expresses the estrogen receptor, metastasizes spontaneously, and responds to antiestrogen treatment, will allow for further investigation into the biology and potential treatment of metastasis.

## 1. Introduction

Breast cancer (BC) is the most common cancer diagnosed in women worldwide after skin cancer, and remains the leading cause of female cancer mortality in 110 countries [1]. Among the different BC subtypes, estrogen receptor-positive (ER+) BC is the most common, and has a more favorable prognosis than triple-negative (TN) BC [2]. ER+ and TN BC also differ in various biological aspects, such as immune responses (ER+ BC is often described as “immune cold”), metastatic tropism, and therapeutic options with an emphasis on antiestrogen treatment for ER+ BC. Despite the large emotional and financial burden ER+ BC places on patients, there is a lack of comprehensive and effective models for understanding and managing ER+ BC.

Currently, metastatic models of ER+ BC rely either on exogenous estrogen administration to encourage tumor spread, injection into the vasculature directly, or the use of immunodeficient mice. Studies of ER+ BC metastasis using exogenous estrogen inadvertently modify the immune system function and bone microenvironment (a clinically important location of metastasis in ER+ BC patients), meaning that it is not accurately representative of the human disease [3]. Orthotopic models of BC that originate in the mammary gland and spontaneously metastasize often utilize TN BC cell lines, and many use genetically modified or immunocompromised mouse strains, which can be difficult to maintain and may not translate well to the human disease [4,5,6,7]. Other murine models that use ER+ BC rely on intra-cardiac injection of tumor cells [3,8,9]. While this results in tumor growth in many tissues, it leaves researchers without the ability to study the initial steps of the metastatic cascade. A recent study demonstrated that bone metastasis occurs after intraductal injection of ER+ ZR-75-1 human xenograft cells [9]; however, the xenograft-based system did not allow for the analysis of the interaction between BC and the immune system. Until researchers have a model of ER+ BC that spontaneously metastasizes from the mammary gland in an immunocompetent environment, the mechanisms driving the initial metastatic cascade, the role of the immune system in tumor progression, and metastatic tropism cannot be fully understood.

Considering the above limitations, we developed a model of ER-expressing BC that spontaneously metastasizes from the mammary gland and is responsive to antiestrogen therapy in immunocompetent mice. We genetically modified two bone-tropic, TN BC cell lines (4T1.2 and E0771/bone) to express ERα. As the ER-expressing 4T1.2-derived tumors showed promising results in tumor translatability, ER-expressing and TN 4T1.2 cells were injected into the mammary glands of syngeneic mice. The metastasis to major organs and long bones, and primary tumor immune infiltration were assessed. To ensure a response to antiestrogen therapies, a subset of mice was then treated with either the selective estrogen receptor modulator (SERM) tamoxifen (TAM) or the selective estrogen receptor degrader ICI 182,780 (ICI). We found that when compared to mice with TN tumors, ER-expressing tumors had modified immune cell infiltration, had smaller lung metastases, and were responsive to antiestrogen therapy. 

## 2. Materials and Methods

### 2.1. Cell Line Generation

The 4T1.2 and E0771/Bone murine bone-tropic, TN BC lines were provided under material transfer agreements with Drs. Hiraga and Anderson, respectively [5,7]. These two sublines were derived from the parental TN 4T1 and E0771 cell lines and have increased bone tropism due to repeated intracardiac injection and isolation of the resulting bone metastatic subclones (thus creating the 4T1.2 and E0771/bone lines, respectively). Using a human, wild-type HEGO ERα-GFP construct developed by Dr. Elaine Alarid, we generated ER-expressing BC sublines of both cell lines in collaboration with the Wake Forest Baptist Comprehensive Cancer Center Cell Engineering Shared Resource (CESR). To confirm its identity, the ERα-GFP construct was sent to GeneWiz (South Plainfield, NJ, USA) for Sanger sequencing using standard M13-F and M13-R sequencing primers. The ERα-GFP expression cassette was cloned into a lentivirus transfer vector using methods based upon the ViraPower™ Lentiviral Gateway™ Expression system (ThermoFisher Scientific, Waltham, MA, USA). Briefly, following sequencing, the ERα-GFP expression cassette was PCR amplified to introduce the attB and attL-R sites using following primers:   Forward: 5′-CACCACGGCCACGGACCATGA-3′
Reverse: 5′-TTACTTGTACAGCTCGTCCATGCCGAG-3′

A Gateway entry clone was generated using the Gateway pENTR/D-TOPO linear vector cloning kit (ThermoFisher Scientific) to insert the ERα-GFP expression cassette into the Gateway donor vector. Restriction analysis using Noti-HF (NEB #R3189L, 20K U/mL, Lot 10030794), NruI-HF (NEB #R3192S, 20K U/mL, Lot 10030601), and AgeI-HF (NEB #R3552S, 20K U/mL, Lot 10028839) was performed with a CutSmart Buffer at 37 °C for 5–15 min or overnight to verify the correct orientation of the insert. The ERα-GFP expression cassette was then transferred from the entry clone to a Gateway compatible lentivirus transfer vector (pLenti CMV Blast DEST (706-1)), a gift from Eric Campeau and Paul Kaufman (Addgene plasmid #17451; http://n2t.net/addgene:17451 (accessed on 11 September 2019); RRID: Addgene_17451, [10]) using LR Clonase II. The transfer vector was then packaged into a lentivirus using the pPACKH1 HIV third generation lentiviral expression system kit and PureFection reagent (both from System Biosciences, Palo Alto, CA, USA) into HEK293-T cells (ATCC, Manassas, VA, USA). Lentiviral particles were concentrated using the Clontech Lenti-X Concentrator (Takara Bio USA, Mountain View, CA, USA). The concentrated lentivirus was used to transduce the cell lines, and ERα-GFP-expressing cells were obtained after antibiotic selection for two weeks. A lentivirus transfer vector expressing GFP alone (pLenti CMV GFP Puro (658-5), a gift from Eric Campeau and Paul Kaufman; Addgene plasmid #17448; http://n2t.net/addgene:17448 (accessed on 11 Septemeber 2019); RRID: Addgene_17448, [10]) was packaged into lentiviral particles as described above. Both TN BC parental cell lines were transfected with this vector and GFP-expressing cells were obtained after antibiotic selection. All cells tested negative for mycoplasma and murine viruses prior to injection into animals.

### 2.2. Estrogen Signaling Gene Array, Immunoblot, and Estrogen Response

ERα signaling genes were compared via PCR between 4T1.2 and E0771/Bone ER-expressing BC and TN BC using the estrogen receptor signaling (SAB target list) 96-well plate from Bio-Rad (Hercules, CA, USA). Cells were grown according the ATCC guidelines (in an estrogen-rich environment) for their parental lines with the inclusion of the appropriate selection antibiotics in 10 cm dishes until 70% confluent. RNA was extracted from cell lysates using the Qiagen RNeasy Micro Kit (Hilden, Germany) according to the manufacturer’s instructions. From the extracted RNA, equal concentrations of cDNA for each sample were created using the iScript cDNA Synthesis Kit from Bio-Rad according to the manufacturer’s instructions. The expression levels of downstream ERα genes were calculated relative to *Gapdh* (*Hprt* and *Tbp* were also used as controls with similar results) and was quantified using SsoAdvanced Universal SYBR green (Bio-Rad) and a CFX Connect Real-Time PCR Detection System (Bio-Rad) using the manufacturer’s recommended cycling parameters.

To evaluate the changes in protein expression of ERα in our ER-expressing 4T1.2 cell line, as well as changes in downstream protein products from ERα activation (Src), protein immunoblotting on the collected cells was performed. Briefly, cells were grown in complete media for 24 h before being collected in RIPA buffer (ThermoFisher, Waltam, MA, USA) on ice. Cellular protein was collected and 20 µg per lane was run on a 4–15% SDS-PAGE gel and transferred to a 0.2 μm nitrocellulose membrane for blotting with antibodies against the proteins of interest (Appendix A). Blots were imaged and densitometry was performed using ImageJ (Version 1.54 g) and normalization was performed by calculating normalization factors based on the b-actin intensity of the blot.

To assess cell responses to estrogen, ER-expressing and TN 4T1.2 and E0771/bone cells were grown in full media (high-estrogen) and in phenol red-free and charcoal stripped FBS (low-estrogen) for 48 h in the Incucyte Zoom (Essen BioScience, Ann Arbor, MI, USA). Confluence at 48 h was calculated and the increased growth in full media compared with the estrogen-low media was calculated (*n* = 5). 

### 2.3. Animal Experiments

All experiments were performed in accordance with the Institutional Animal Care and Use Committee at the Wake Forest University School of Medicine (IACUC A20-044). To characterize the primary tumors and examine the distribution of metastases, ER-expressing (*n* = 25) and TN (*n* = 25) 4T1.2 BC cells were implanted in seven-week-old, intact female, Balb/c mice obtained from Charles River Laboratories (Wilmington, MA, USA) and housed with 12 h dark/light cycles and ad libitum access to food and water. Animal numbers were based on a power calculation using the SAS software (Version 9.4M7) such that 80% power would be achieved with 23 animals to detect the difference in the hind limb metastasis rate based off of previous internal studies that indicate a 25% metastasis rate to the hind limb in ER-expressing tumor-bearing mice. Mice were restrained and 1 × 10^5^ ER-expressing or TN BC cells suspended in 20 µL of sterile phosphate-buffered saline (PBS) were injected into the 4th inguinal mammary fat pad. Tumor volume was measured every three days for five weeks or until humane endpoints were met. Tumors, visceral organs, and long bones were collected for histology. Tumors and visceral organs were weighed and either fresh-frozen or fixed in 4% paraformaldehyde prior to being embedded in paraffin. After fixation in 4% paraformaldehyde, the hind limb bones were isolated from whole legs and decalcified in 14% neutral buffered EDTA for two weeks prior to being embedded in paraffin.

To elucidate the responsiveness of the tumors to antiestrogen treatment, seven-week-old, intact female, Balb/c (*n* = 21) and C57BL/6 (*n* = 27) mice from Charles River were similarly housed. Mice were restrained and 1 × 10^5^ ER-expressing or TN 4T1.2 or E0771/Bone cells suspended in 20 µL of PBS were injected into the 4th inguinal mammary fat pad. Tumors were established and reached a volume of 100 mm^3^. At this time, mice with ERα-expressing tumors were treated with either TAM at 5 mg over 60 days as a time-release pellet (ER-expressing 4T1.2, *n* = 6; ER-expressing E0771/Bone, *n* = 7), ICI at 1 mg per week subcutaneously suspended in castor oil and ethanol (ER-expressing 4T1.2, *n* = 6; ER-expressing E0771/Bone, *n* = 7), or a subcutaneous injection of castor oil and ethanol as a vehicle control (ER-expressing 4T1.2, *n* = 4; ER-expressing E0771/Bone, *n* = 6). Tumor volumes were measured every three days. After 21 days of treatment, the mice were humanely euthanized and tumors, visceral organs, and long bones were collected. Tumors were weighed and fresh-frozen or fixed in 4% paraformaldehyde prior to being embedded in paraffin. After fixation in 4% paraformaldehyde, the hind limb bones were isolated from whole legs and decalcified in 14% neutral buffered EDTA for two weeks prior to being embedded in paraffin.

### 2.4. Immunofluorescence, Histology, and Immunohistochemistry

Immunofluorescence was performed on transduced cell lines for ER (Appendix A). Briefly, ER-expressing or TN 4T1.2 or E0771/Bone cells were grown on a chambered slide at 1 × 10^6^ cells/well. After 24 h of growth on the slide, the cells were fixed with 10% neutral buffered formalin. The cells were permeabilized and blocked with 3% bovine serum albumin (BSA). The cells were incubated with primary antibody or blocking buffer as a negative control for one hour at room temperature. The slides were washed in PBS, and then incubated with a secondary antibody (Appendix A). Coverslips were mounted with antifade mounting media with DAPI.

Hematoxylin and eosin (H&E) staining was used to identify and quantify metastatic lesions in formalin-fixed, paraffin-embedded sections of major organs including the brain, lungs, heart, liver, and kidneys. Immunohistochemistry on formalin-fixed, paraffin-embedded tumors and hind limbs were performed to characterize the tumors, identify metastatic tumor cells, and to profile the bone microenvironment. Paraffin-embedded tumors and bones were sectioned into 5 µm thick sections and placed on charged glass slides. For tumors, antigen unmasking was performed by heat-induced epitope retrieval using 0.05% citraconic anhydride solution (pH 7.4) for 45 min at 98 °C; for bone samples, antigen unmasking was performed by heat-induced epitope retrieval using Tris-EDTA PH 9.0 for 20 min at 95 °C. Endogenous horseradish peroxidase (HRP) activity was quenched by incubating with BLOXALL blocking solution for 10 min. The samples were blocked with 1% BSA for 30 min at room temperature then incubated with primary antibody overnight at 4 °C. Immunohistochemistry of the tissue sections was performed using antibodies against ER, CD3, CD4, CD8a, CD68, CD45r, and neutrophil elastase on the primary tumors and pan-cytokeratin, Sca-1, and endomucin on the bone (Appendix A). After incubation, the samples were washed with PBS and then incubated with an HRP-conjugated secondary antibody (Vector Laboratories, Burlingame, CA, USA) followed by Nova Red chromogen (Vector Laboratories, Burlingame, CA, USA) for staining development. All samples were counterstained with hematoxylin. Slides were then scanned using a Hamamatsu NanoZoomer by the Virtual Microscopy Core in the Wake Forest University School of Medicine or using an Olympus VS120 Slide Scanner (Olympus, Tokyo, Japan). Immunostaining was quantified using the VisioPharm digital pathology analysis software (Version 2018.4). Briefly, a total area of region of interest (ROI) was measured for each specimen; custom-designed apps were then used to identify and measure the regions with positive staining within the ROI, and the ratios of the positive staining area vs. the total area were calculated. Three separate, non-consecutive tissue sections of whole tibiae (including trabecular regions in the metaphysis, the epiphysis, and the diaphysis) were analyzed for metastases with H&E staining for the initial fifty mice. Sections of lung and hind limbs were scanned using the Olympus VS120 Slide Scanner and metastases were outlined using the VisioPharm digital pathology software (Version 2018.4) to calculate the area of metastases. For the mice treated with antiestrogen therapies, pan-cytokeratin immunohistochemistry was performed on three non-consecutive tibial sections to quantify the number of metastases.

### 2.5. Bone Structure and Osteoclast Analyses

To visualize the macroscopic changes in hind limb bone structure, radiographs of fresh hind limbs cleaned of skin and skeletal muscle were taken using the Faxitron Multifocus 10 × 15 Digital Radiography System (Hologic, Marlborough, MA, USA). Images were acquired with polyester and aluminum cubits as standards with kV and time automatically calculated for optimal image quality. Radiographs were analyzed using ImageJ (version 1.8.0_112; 64-bit; National Institute of Health) in accordance with previously published methods [11]. Briefly, images were scaled to 8-bit images with the minimum and maximum pixel intensity manually set in accordance with the polyester and aluminum cubits. The images were inverted and pseudo-colored with a 16-color scale. Soft tissue remnants adjacent to bone were removed using Photoshop. Relative bone density was analyzed using the frequency of pixel hue and the data were analyzed as the cumulative frequency of pixel intensity within the bone.

To visualize microscopic changes in the bone structure and osteoclasts, paraffin-embedded bones were sectioned at 5 µm and placed on charged glass slides. The sections were stained for one hour at 37 °C in tartrate-resistant acid phosphatase (TRAP) staining solution (0.3 mg/mL Fast Red Violet LB, 0.05 M sodium acetate, 0.03 M sodium tartrate, 0.05 M acetic acid, 0.1 mg/mL naphthol, 0.1% Triton X-100, pH 5.0) to visualize osteoclasts. The sections were counterstained with Gill Method Hematoxylin Stain 1 (ThermoFisher). The slides were scanned at 20× with a Hamamatsu NanoZoomer by the Virtual Microscopy Core at the Wake Forest University School of Medicine or with an Olympus VS120 Slide Scanner. Bone histomorphometry, osteoclast numbers, and growth plate organization were analyzed using the BioQuant Osteo software (Version 2016v16.1.60; RRID: SCR_016423) using a ROI 150 µm distal to the proximal physis encompassing the trabecular region of the tibiae with an area of 1500 µm. Osteoclasts were detected based on thresholding for the red/purple TRAP staining. The measurements generated included bone volume normalized by tissue volume, bone fraction, bone surface normalized by bone volume, and number of osteoclasts per millimeter of bone surface. Trabecular thickness was calculated using trabecular thickness = 2/(bone volume/bone surface) × 1000.

### 2.6. Statistical Analysis

Normalcy was tested using a Shapiro–Wilk test. To determine statistical significance, a Student’s *t*-test (if normally distributed), a Mann–Whitney (if data did not pass normalcy), Wilcoxon test, one-way, or two-way analysis of variance (ANOVA) with Tukey post-test was used to the analyze data using the GraphPad Prism 9 software (RRID: SCR_002798). Comparisons of metastatic organ tropism between ER-expressing BC and TN BC cells were performed using a Chi-squared test with a Fisher’s exact test used to calculate *p* values. Error bars represent the standard error of the mean (SEM). * represents *p* < 0.05, ** represents *p* < 0.01, and *** represents *p* < 0.005.

## 3. Results

### 3.1. ER-Expressing BC Expressed Higher Levels of ER Signaling Genes and More ER Protein

To confirm the appropriate function of the ERα construct inserted into the TN BC cell lines, PCR arrays for downstream genes regulating ERα expression and products of downstream ERα signaling were performed. Of the 79 genes detected in the array, 63 were upregulated (79.7%) in the ER-expressing 4T1.2 cell line compared with TN 4T1.2 (Figure 1). Notably, the *Esr1* gene, which regulates ERα expression [12], was upregulated 17.39-fold in the ER-expressing 4T1.2 cell line compared with the TN 4T1.2 cell line. Other genes of note included *Bdnf* (33.59-fold increase), *Ptgs2* (19.42-fold increase), *Ctgf* (16.00-fold increase), and *B2m* (12.55-fold increase). The ER-expressing E0771/bone cell line showed comparable results with 77.6% of detected genes upregulated and with *Esr1* upregulated 4.59-fold compared with the TN E0771/bone cell line (Appendix A). Other ERα-responsive genes that were upregulated in the E0771/bone ER-expressing BC line included *Wisp2* (7.11-fold increase), *Ltbp1* (6.10-fold increase), *Bcl2* (5.02-fold increase), and *Xbp1* (4.43-fold increase) (Appendix A).

By immunofluorescence, the cell lines transduced to express ER had higher ER immunoreactivity than the TN cell lines (Figure 1 and Appendix A). By immunoblotting, the ER-expressing 4T1.2 cell line had increased expression of phospho-ERα, ERα, and Src (a downstream protein of ERα activation; Figure 1, full blot with densitometry in Appendix A). ER-expressing 4T1.2 cells were significantly more confluent at 48 h of exposure to complete media than TN 4T1.2 when normalized to low estrogen media at the same time point (Appendix A).

### 3.2. ER-Expressing and TN 4T1.2 Tumors Grow Similarly, Have Similar Metastatic Niches, and Both Metastasize to the Hind Limbs

Tumors from the mice injected with ER-expressing and TN 4T1.2 cells grew at similar rates and had similar end weights (average final tumor weight: TN = 1.04 g, ER-expressing = 0.93 g, *p* > 0.05, Wilcoxon test, Student’s *t*-test; Figure 2). The total ERα expression in the ER-expressing tumors was significantly higher than in the TN tumors, as analyzed by IHC (*p* < 0.05, Mann–Whitney test; Appendix A). The average final body weight was not significantly different between groups (ER-expressing = 21.50 g ± 0.35, TN = 20.79 g ± 0.23, *p* > 0.05, Student’s *t*-test). Organ weights in grams and organ weight as a percentage of body weight were not significantly different between the mice with ER-expressing and TN 4T1.2 tumors (Appendix A). The average splenic weights (ER-expressing = 0.60 g ± 0.04, TN = 0.74 g ± 0.08) for both groups were markedly elevated over age- and strain-matched control data provided by Jackson Labs (average splenic weights are 0.082 g and 0.41% body weight). Like the organ weight data, there were no significant differences in which major organs had histologic evidence of metastases as determined by a board-certified veterinary anatomic pathologist (*p* > 0.05, Chi-squared test, Table 1; examples of H&E in major organs included in Appendix A). The highest incidence of metastases for both groups was in the lungs. While the number of metastases per histologic section was not different between groups, the average area of the tumors was significantly smaller in the ER-expressing group (19,875 µm^2^) than in the TN group (70,905 µm^2^; *p* < 0.05, Mann–Whitney test, Figure 2). Importantly, both groups developed metastases within the hind limbs as assessed via three non-consecutive H&E-stained microscopic sections. The prevalence of hind limb metastases in the ER-expressing group was 8% (2 of 24 animals) and in the TN group, it was 24% (5 of 21 animals).

### 3.3. ER-Expressing 4T1.2 Tumor Immune Infiltration was Modified Compared with TN 4T1.2 Tumors

ER expression in patients with BC is associated with decreased immune cell infiltration and/or activity within the primary tumor, although the underlying mechanisms are poorly understood [13]. Furthermore, immune infiltration within the tumor microenvironment plays a critical role in the development and progression of BC [14]. Therefore, describing the immune landscape of the ER-expressing and TN 4T1.2 tumors was critical for ensuring this orthotopic BC model is representative of the natural disease state. To this end, we utilized immunohistochemistry with custom-made VisioPharm (Version 2018.4) analysis apps to determine the percent of total tumor area with immunoreactivity to CD68, CD3, CD4, CD8a, neutrophil elastase, and CD45r. Examples of the staining are shown in Figure 3A. ER-expressing 4T1.2 tumors were associated with significantly increased CD68 macrophage expression (ER-expressing = 0.24% total area ± 0.08, TN = 0.07% total area ± 0.02, *p* < 0.05, *t*-test) and significantly increased CD3+ T cell expression (ER-expressing = 1.28% total area ± 0.35, TN = 0.22% total area ± 0.04, *p* < 0.05, Student’s *t*-test; Figure 3B). Although general T cell expression was increased in ER-expressing tumors, the expression of CD4+ and CD8a+ T cells were both significantly decreased in ER-expressing (0.04% total area ± 0.01 and 0.003% total area ± 0.0007, respectively) compared with TN tumors (0.82% total area ± 0.20 and 0.016% total area ± 0.004, respectively) (*p* < 0.05, Student’s *t*-test). There was no significant difference in CD45r or neutrophil elastase positivity.

### 3.4. There Are No Significant Differences in Macro- or Microscopic Changes to the Hind Limb in Mice with ER-Expressing or TN BC

Bone metastases from BC are a serious complication and patients with ER+ BC are at an increased risk for developing bone metastases which can lead to osteoblastic, osteolytic, or mixed reactions with osteolytic being the most common [15]. Furthermore, with primary breast cancer, the bone niche is modified prior to metastases arriving in the tissue, including increased vasculature growth and hematopoietic stem cells [7,16,17,18,19,20]. To assess whether there were modifications in the pre-metastatic niche in these mice or if bone metastases in these mice were associated with gross bone changes, radiographs of the hind limbs were taken, and the mineral density was mapped. No significant difference in bone density was detected between mice harboring ER-expressing and TN tumors (Appendix A). To assess the modifications within the bone microenvironment, histomorphometry of the tibia just distal to the proximal growth plate was used (Figure 4A) to determine the bone volume to tissue volume fraction. No significant changes in the osteoclast surface-to-bone surface ratio, osteoclast count, bone volume-to-tissue volume ratio, trabecular count, or trabecular thickness were found between mice with ER-expressing and TN BC tumors without bone metastasis (Figure 4B–F). Furthermore, in mice with bone metastases, there were no differences between the TN and ER-expressing tumors, or between mice with and without bone metastases. 

### 3.5. ER-Expressing Tumors Were Responsive to Antiestrogen Treatments

To examine the effects of antiestrogen treatment on tumor growth, ER-expressing and TN 4T1.2 or E0771/Bone cells were injected into Balb/c or C57BL/6 mammary fat pads, respectively. The average tumor volumes were significantly elevated (approximately 1.5-fold increase) for ER-expressing 4T1.2 (900 mm^3^)- and ER-expressing E0771/Bone (1677 mm^3^)-derived tumors when compared with the tumors derived from the TN parental cell lines (658 mm^3^ and 1029 mm^3^, respectively; two-way ANOVA, *p* < 0.05, Figure 5). At study termination, the average tumor weight in mice injected with ER-expressing E0771/Bone cells (2.44 g) was significantly heavier (1.7-fold increase) than in those injected with TN E0771/Bone cells (1.17 g; Appendix A, one-way ANOVA, *p* < 0.05). Additionally, ER-expressing 4T1.2 tumors weighed more (1.4-fold increase; 0.98 g average) than TN 4T1.2 tumors (0.69 g average; Figure 5, one-way ANOVA, *p* < 0.05). All groups with ER-expressing tumors that were treated with either TAM or ICI had significantly lower tumor volumes and weights than ER-expressing tumors without treatment (two-way ANOVA, *p* < 0.05, Figure 5), except for ER-expressing E0771/Bone tumors treated with TAM which tended to have lower tumor volumes and weights, but the decrease was not significant (two-way ANOVA, *p* < 0.05, Appendix A). ER expression in the primary tumor, as determined by IHC, was significantly decreased in mice treated with ICI (a selective estrogen receptor degrader) compared with the ER+ tumors without treatment (*p* < 0.05, Mann–Whitney Test; Appendix A). Mice treated with TAM (a selective estrogen receptor modulator, with agonist/antagonist properties) did not have a significantly different ER expression level compared with the non-treated, ER-expressing mice (*p* > 0.05, Mann–Whitney Test; Appendix A).

### 3.6. ERα-Expressing 4T1.2 Tumors in the Bone Were Not Responsive to Antiestrogens

To further examine the response to antiestrogens, hind limbs were isolated from mice injected with ER-expressing BC after treatment with TAM (*n* = 6) or ICI (*n* = 6) or vehicle control (*n* = 4) and compared to hind limbs from TN BC-injected mice (*n* = 5). The tibiae of the mice were examined with immunohistochemistry for anti-pan-cytokeratin, which would identify the TN and ER-expressing epithelial tumor cells injected into the mice. Only mice injected with ER-expressing 4T1.2 cells had visible metastases within the bone in this cohort of mice (Figure 6A). Interestingly, metastases were observed in all ER-expressing 4T1.2 groups (1 of 4 for vehicle control, 2 of 6 for TAM, and 1 of 6 mice for ICI had hindlimb metastases), regardless of treatment (Figure 6B), likely due to the dissemination of tumor cells from the established tumors prior to the initiation of treatment.

Disseminated BC cells enter the bone microenvironment through the vasculature and colonize the bone at the hematopoietic stem cell (HSC) niche [10]. To assess whether changes in the colonization niche were different between tumor types, we stained bones for Sca-1+ HSC cells and endomucin+ blood vessels (Appendix A). We found that mice with ER-expressing E0771/Bone-derived tumors had significantly increased percentages of Sca-1-positive HSCs (64.4-fold increase) and endomucin-positive vasculature (2.8-fold increase) within the bone marrow when compared with TN E0771/Bone tumors (Appendix A). There were no significant differences between Sca-1 or endomucin positivity between mice with ER-expressing or TN 4T1.2-derived tumors (Student’s *t*-test, *p* > 0.05, Appendix A).

### 3.7. TAM Induced Alterations in the Bone Niche

To further characterize the alterations in the bone microenvironment based on tumor type, we examined the bone structure and osteoclast differentiation. Histomorphometry on the tibiae just distal to the growth plate was used to determine the bone volume-to-tissue volume fraction. No significant changes in the bone fraction were measured between mice with ER-expressing and TN BC tumors (Figure 7). Mice injected with ER-expressing 4T1.2 cells that were treated with TAM had a significantly increased bone volume-to-tissue volume percentage when compared with ER-expressing 4T1.2 cell-injected mice that were treated with ICI (two-way ANOVA, *p* < 0.01, Figure 7). Similarly, mice injected with ER-expressing E0771/Bone cells that were treated with TAM tended to have an increased bone volume-to-tissue volume percentage when compared with all other groups (Appendix A). Thus, TAM treatment alone may alter the bone structure in ways that could alter metastatic spread.

BC bone metastases are often osteolytic, and the activation of osteoclasts is associated with the reactivation of dormant BC cells in the bone [21]. TRAP+ osteoclasts were quantified in mice implanted with ER-expressing and TN BC. No significant changes in osteoclast number were seen between ER-expressing and TN BC-injected mice (Figure 7). ER-expressing E0771/Bone tumors treated with TAM had a significantly increased percentage of osteoclasts at the bone surface over total bone surface when compared with both untreated ER-expressing E0771/Bone mice and with the parental E0771/Bone mice (two-way ANOVA, *p* < 0.05, Appendix A). There were no significant differences in the osteoclast surface-to-bone surface percentage in any of the mice with 4T1.2 tumors (Figure 7). Thus, neither osteoclast activation nor alterations in bone structure were responsible for the increased ER-expressing 4T1.2 bone metastasis.

## 4. Discussion

An ER-expressing BC mouse model that spontaneously metastasizes from the mammary tissue without exogenous estrogen treatment will allow researchers to study the role of ER signaling in BC metastasis in a biologically relevant model. Additionally, this allows for the study of ER signaling within the tumor and in the tumor microenvironment, facilitating the study of antiestrogen therapies, and providing a platform for a preclinical model for drug development. While our E0771/bone cell line has evidence of increased estrogen expression, it lacked metastatic tropism to important tissues and was therefore not the focus of the main body of work. Our ER-expressing 4T1.2 mouse model shared clinically relevant features with the natural disease, and spontaneous metastasis developed without the administration of exogenous estrogen in an immunocompetent host. 

Our 4T1.2 tumor-bearing mouse model of ER-expressing BC exhibited similarities to the human disease including the upregulation of genes downstream of ER activation, being immunologically cold, exhibiting smaller metastases in the lung (possibly indicating lower aggressiveness), metastasis to bones, and response to antiestrogen therapies. The upregulation of the estrogen-responsive genes and protein product (src), and phospho-ERα and ERα following lentiviral transduction indicates a functional ER in these cell lines. The ER-expressing 4T1.2 primary tumors showed modified immune infiltration that in many ways recapitulated what is seen in humans, further exemplifying the functionality of the transduced ER. Interestingly, while our ER-expressing E0771/bone cell line had increased expression of ER-regulated genes, once placed in an animal model, the cell line failed to exhibit bone metastases, which is an important characteristic of ER+ BC in humans. 

In humans, ER+ BC is clinically less aggressive than TN BC [22,23], as demonstrated by longer patient survival times, better responses to treatment, and fewer metastases [22]. In our study, ER-expressing 4T1.2-derived tumors grew at the same rate and had similar final volumes as TN 4T1.2-derived tumors. Although the incidence of metastases by organ was not different between the ER-expressing and TN tumor-bearing mice, the size of the tumors in the lung (which was the primary organ of metastasis in this model) was significantly smaller in the ER-expressing tumor-bearing mice. This likely represents less aggressive tumor metastases in the ER-expressing model, either through a longer time to metastasis initiation or through decreased metastatic growth in the end-organ.

One purpose of our study was to develop an ER-expressing BC cell line with the capability to metastasize to bones. While we did develop an ER-expressing tumor line that spontaneously metastasized to bones, only mice with tumors from 4T1.2 cells, but not the E0771/bone cells, developed metastases to the bone. Bone metastases in the ER-expressing and TN 4T1.2 tumor-bearing mice were present by 4 to 5 weeks post injection in the mammary glands. This short time course for bone metastasis development is similar to the time course described in the seminal intra-mammary model of 4T1.2 cells, where cancer cells with stem-like properties were identified as early as 22 days following intra-mammary injection [24]. Another model of intra-mammary injection of tumor cells with bone metastases found that bone metastasis did not occur within the first 30 days following the injection of TN 4T1 tumor cells [25]. This may be due to differences in detection methods, as this study used a luciferase reporting system to identify metastases which may not be able to detect metastases if the cell numbers are small. Alternatively, it may represent differences in bone tropism and the upregulation of metastatic mechanisms in the 4T1.2 cell line. 

It is important to note that while our mice had an 8% incidence of hind limb metastasis in the ER-expressing 4T1.2 group and a 24% incidence in the TN 4T1.2 group, the antiestrogen administration experiment found a 25% incidence in the ER-expressing group and 0% incidence in the TN 4T1.2 group. This study was performed in a separate vivarium at our institution which has repeatedly found a 25% incidence of hind limb metastasis in ER-expressing 4T1.2 tumor-bearing mice (*n* = 180). Furthermore, the experiments that found the tumor growth and weight differences between the mice in Figure 2 and Figure 5 were performed in different vivaria, which likely influenced the differences between growth rates. The cells used in both vivaria were from the same lentiviral transduction at a low passage number. The reason for the discrepancy between these outcomes is unclear, although it is repeatable. Differences beyond our control, such as environmental stress, slight differences in the handling of cells and mice, or intrinsic differences in the mice, may be at play. While interesting, in both scenarios, ER-expressing 4T1.2 tumor cells injected orthotopically spontaneously metastasized to the bone without estrogen supplementation in an immunocompetent host.

Immune infiltration within the tumor microenvironment plays a critical role in the development and progression of BC [14]. ER expression is associated with decreased immune cell infiltration and activity within the primary tumor, although the mechanisms by which this is accomplished are poorly understood [13]. It is important to note that the TN 4T1.2 cell line was transduced to express GFP and the ER-expressing 4T1.2 cell line expressed ERα-GFP, and therefore, the difference is unlikely to be due to the presence of GFP alone. In our study, ER-expressing 4T1.2 tumors had different intra-tumoral immune profiles, with significantly increased counts for CD3+ T cells and significantly decreased CD4+ and CD8a+ T cell counts. Decreased numbers of tumor-infiltrating lymphocytes have been observed in patients with ER+ disease when compared with TN tumors [26,27,28,29,30,31]. While CD4+ and CD8+ T cells were significantly decreased in the ER-expressing 4T1.2-derived tumors, CD3+ T cells (which is a general marker of T cells) were significantly increased compared with the TN 4T1.2-derived tumors. The CD4- and CD8-negative population of T cells that accounts for this increase is unclear but may represent an infiltrating double-negative population. Double-negative CD3+ T cells in cancer are hypothesized to play an antitumor effect with immune regulatory functions. While this population of double-negative T cells may represent a large proportion of the infiltrating T cells in the ER-expressing 4T1.2-derived tumors, in humans, this population of T cells represents only up to 5% of circulating T cells [32]. Further studies on the specific T cell population and their role in tumor progression are needed. 

Aside from the influence of T cell infiltration into the mammary tumor, the influence of other immune cells within the tumor microenvironment has a substantial impact on patient prognoses. For example, tumor-associated macrophages are associated with a worse prognosis [33]. The ER-expressing 4T1.2 tumors trended toward increased CD68+ macrophages compared with the TN 4T1.2 tumors. While unexplored, it may be possible that this population of macrophages in the ER+ tumors may be skewed toward M2 polarization, which could account for the decreased CD4+ and CD8+ T cell infiltration [34]. In patients with ER+ BC, there is decreased expression of M1-specific transcripts within primary tumors compared with TN BC according to both the TCGS and METABRIC repositories [19], possibly indicating an increased M2 polarization. While assessing long-term disease progression was outside of the scope of our study, it is plausible that the differing immune landscapes seen in ER+ tumors when compared with TN tumors may have implications for tumor progression.

Interestingly, when given antiestrogens, at least one mouse per treatment group with ER-expressing 4T1.2 BC had evidence of bone metastasis. It is possible that these metastases occurred before the antiestrogen therapy began, since tumors were allowed to establish and grow to 100 mm^3^ before treatment began. One of the antiestrogen compounds used in our study, TAM, is a SERM and is one of the most frequently used treatments for ER+ BC [30]. As a SERM, TAM works in an antiestrogen manner in the tumor to inhibit tumor growth, while acting in a pro-estrogen manner in the bone to maintain bone volume in post-menopausal women [7,20]. The results reported here are consistent with previous studies in which mice treated with TAM had increased bone volume when compared with mice treated with a vehicle control. While the mice with TN tumors in this study were not treated with antiestrogen compounds, in a separate study, we treated non-tumor-bearing mice with TAM and noted similar increases in bone volume. Future studies are warranted to test other breast cancer drugs and drug combinations on the development and progression of bone metastases derived from ER-expressing 4T1.2 cells. 

Previous studies across multiple cancer types have demonstrated that tumors can modulate the pre-metastatic bone microenvironment to become a more hospitable environment by increasing HSC numbers and vascularization [16,17,18]. We did not find changes in the bone microenvironment between mice with ER-expressing and TN 4T1.2 tumors, which were the only cell lines in this study to exhibit bone metastases. This suggests that the 4T1.2 cell line does not modify HSCs or the vasculature in the pre-metastatic niche. The percentages of Sca-1- and endomucin-positive cells, markers for HSC/tumorigenic cells and endothelial/HSCs, respectively, within the bones of mice with ER-expressing E0771/bone-derived tumors were significantly increased when compared with TN E0771/bone-derived tumors. While the specific mechanisms driving the increase in HSCs and endothelial cells is unclear in our study, it is apparent that the expression of ER within the E0771/bone tumors, but not the 4T1.2, mammary tumors, were associated with downstream modification of the bone microenvironment.

While this model may contribute to significant advancements in the detection and treatment of ER+ BC bone metastasis in humans, this study has several limitations. Relatively few mice with ER+ BC developed bone metastasis and estrous cycles were not controlled or monitored in the mice. Additionally, since monitoring for bone metastasis was performed by histological and immunohistochemical evaluation of the tibia, it is possible that some bone metastases were missed in the other long bones or axial skeleton. Future studies will include the introduction of luciferase into the cells. The initial characterization was performed without luciferase as its introduction into 4T1.2 cells was shown to reduce metastatic spread due to interaction with the immune system [35]. Other future studies should also include in vivo exogenous estrogen stimulation with the ER-expressing 4T1.2 tumors to assess the changes in tumor growth and metastasis in this mouse model. Furthermore, only ERα was assessed in this model system as ESR1 was transduced into the cell line. While beyond the scope of this work, future studies to better understand the involvement of Erβ in these cells is warranted to obtain a fuller understanding of the model system. Lastly, mice with TN tumors were not treated with antiestrogen compounds. While we would not expect tumor volumes and weights in the TN group to change since there is no functioning ER, these treatments might alter the bone microenvironment and could affect metastasis and immune cell infiltration.

## 5. Conclusions

In summary and conclusion, we generated an ER-expressing model of BC that spontaneously metastasizes in an immunocompetent host without exogenous estrogen supplementation. This model allows for further exploration of the role of ER in BC tumor progression and metastasis. The ER-expressing 4T1.2 tumor model, which exemplifies important features of ER+ BC bone metastasis in humans, may allow for further investigation into the initial steps and potential treatment of ER+ BC metastasis.

## Figures and Tables

**Figure 1 cancers-15-05773-f001:**
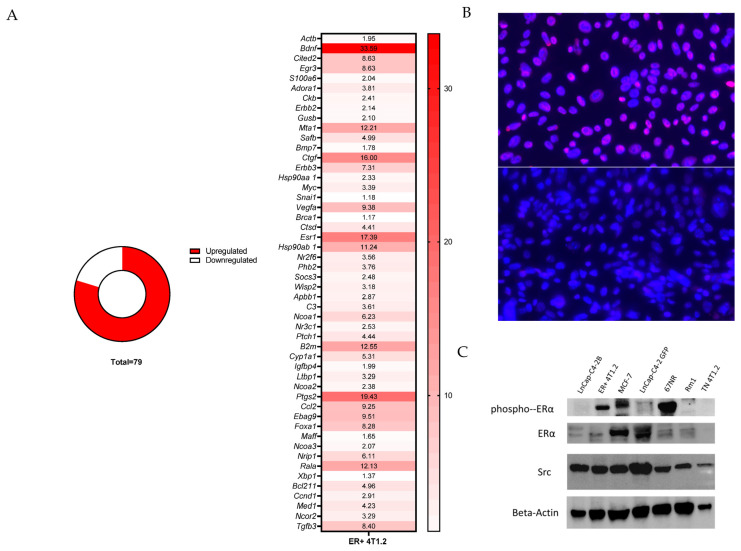
ER-expressing 4T1.2 cells had increased expression of genes and proteins associated with estrogen signaling compared with TN 4T1.2 cells. (**A**) ER-expressing and TN 4T1.2 BC cells were grown according to ATCC guidelines for 4T1 cells and RNA was collected when the cells reached 70% confluence. PCR was performed and the expression levels were normalized to the housekeeping gene *Gapdh* (*Hprt* and *Tbp* were also used for normalization with comparable results) and compared between ER-expressing and TN 4T1.2 cells. Of the 74 genes identified on the PCR array, 63 were upregulated in ER-expressing 4T1.2 cells compared with TN 4T1.2 cells. The upregulated genes are expressed as fold change of ER-expressing over TN. (**B**) There was more ER expression (red) in ER-expressing 4T1.2 cells (top) compared with TN 4T1.2 cells (bottom). DAPI was used as a nuclear stain (blue). Images were taken at 10×. (**C**) Immunohistochemistry blots with ER-expressing 4T1.2 and 4T1.2 cell lysate with 67NR (non-metastatic line isogenic to 4T1), 4T1.2 MCF7 (ER+ BC positive control), and LnCap-C42b, LnCap, and Rm1 (negative controls). The uncropped blots are shown in Appendix A.

**Figure 2 cancers-15-05773-f002:**
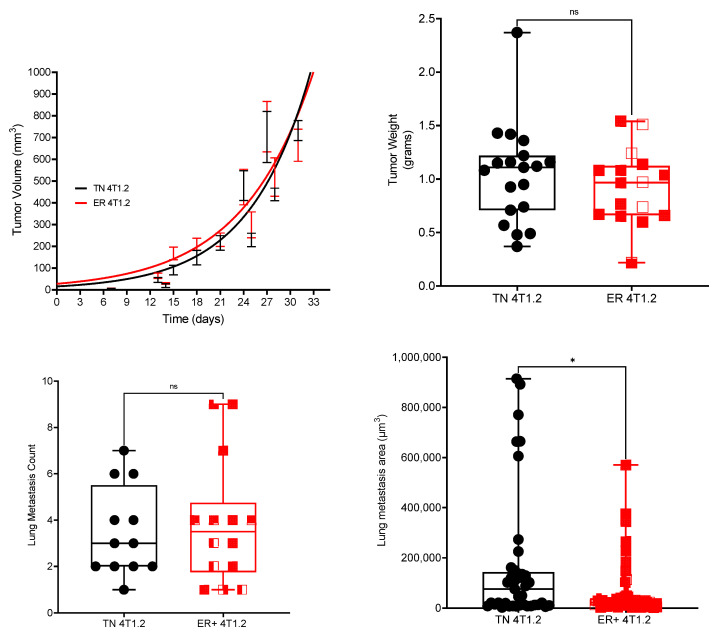
While ER-expressing and TN 4T1.2 tumors grew at similar rates and had similar end weights, the size of the lung metastases was smaller in ER-expressing tumors. ER-expressing and TN 4T1.2 cells were injected into the 4th inguinal mammary fat pad in 25 mice per group. Tumors were measured every 2–3 days and tumor volume was calculated as length × width × width/2 for five weeks or until humane endpoints were met. Tumor growth is plotted as mean ± SEM with a logistic growth curve of best fit and the growth curves were not significantly different (*p* > 0.05). Final tumor weight was measured in grams and was not significantly different between the ER-expressing and TN 4T1.2 tumors (*p* > 0.05, Student’s *t*-test). Lung metastasis count via histologic analysis was not significantly different between TN 4T1.2 and ER-expressing 4T1.2 tumor-bearing mice (*p* > 0.05, Mann–Whitney test). Area of individual metastases in the lung was significantly smaller in the ER-expressing 4T1.2 tumor-bearing mice when compared with those bearing TN 4T1.2 tumors (*p* < 0.05, Mann–Whitney test); ns: no significance. * represents *p* < 0.05.

**Figure 3 cancers-15-05773-f003:**
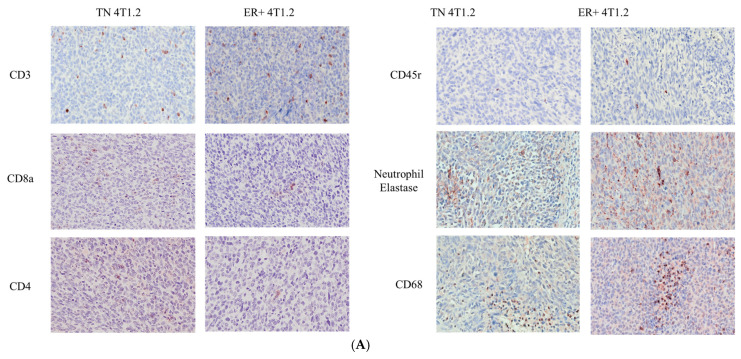
ER-expressing primary tumors have a modulated immune landscape compared with TN tumors. (**A**) Representative images of immunohistochemistry of primary TN and ER-expressing 4T1.2 tumors (images at 20×). (**B**) Quantification of tumors for CD68, CD3, CD4, CD8a, CD45r, and neutrophil elastase was performed using the VisioPharm pathology analysis software (Version 2018.4). The percentage of immunoreactive cells out of the total tumor area was determined with a custom-made app. Percentages of positive cells are represented as single values with the mean ± SEM for the ER-expressing and TN 4T1.2 tumors (*n* = 12–20). * represents *p* < 0.05 and ** represents *p* < 0.005 according to Mann–Whitney test, ns: no significance.

**Figure 4 cancers-15-05773-f004:**
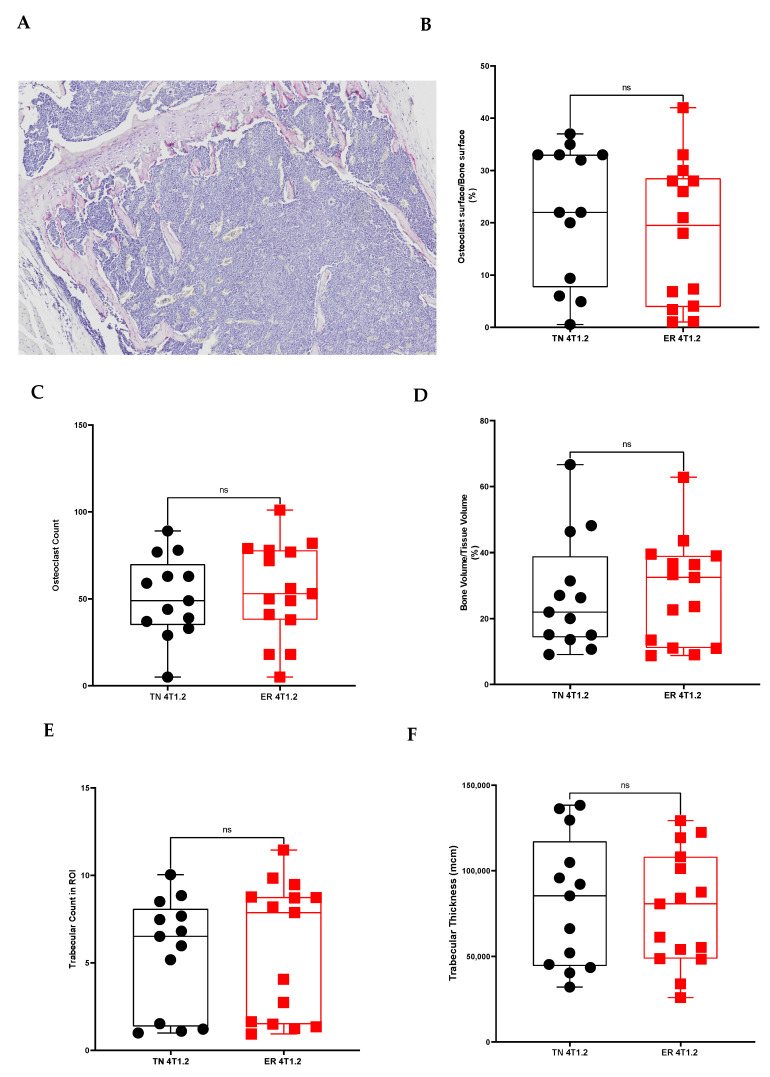
No histological evidence of bone changes was present in mice with ER-expressing or TN 4T1.2 tumors. Tibiae were isolated from mice after tumor implantation with ER-expressing or TN 4T1.2 BC cells. Bones were stained for TRAP+ osteoclasts (**A**; **image at 5×**) and bone histomorphometry was performed to calculate osteoclast surface per bone surface (**B**), osteoclast number (**C**), bone volume-to-tissue volume ratio (**D**), trabecular count per ROI (**E**), and trabecular thickness (**F**). Data are presented as mean percentage ± SEM (*n* = 7), *p* value was calculated using Mann–Whitney test and Student’s *t*-test with significance set at *p* < 0.05, ns: no significance.

**Figure 5 cancers-15-05773-f005:**
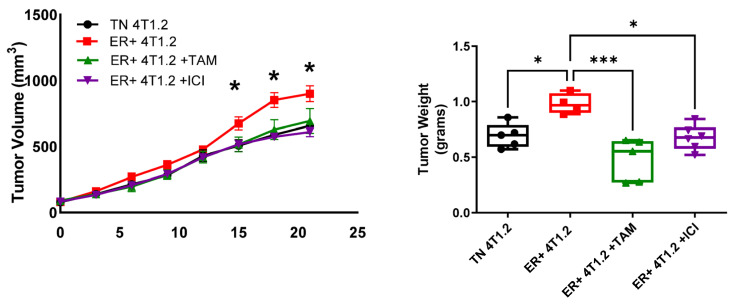
ER-expressing tumors are responsive to antiestrogen treatment. ER-expressing and TN 4T1.2 BC cells were injected into the 4th inguinal mammary fat pad of Balb/c mice. Once the tumor volume reached 100 mm^3^, the mice were treated with vehicle control, 5 mg/60-day time-release pellet TAM, or 1 mg/wk ICI. Tumor volume was tracked over time and are presented as mean tumor volume ± SEM (*n* = 4–7). Tumor weight was measured upon experimental termination and are presented as mean tumor weight ± SEM (*n* = 4–7). * represents *p* < 0.05 and *** represents *p* < 0.005 according to two-way or one-way ANOVA.

**Figure 6 cancers-15-05773-f006:**
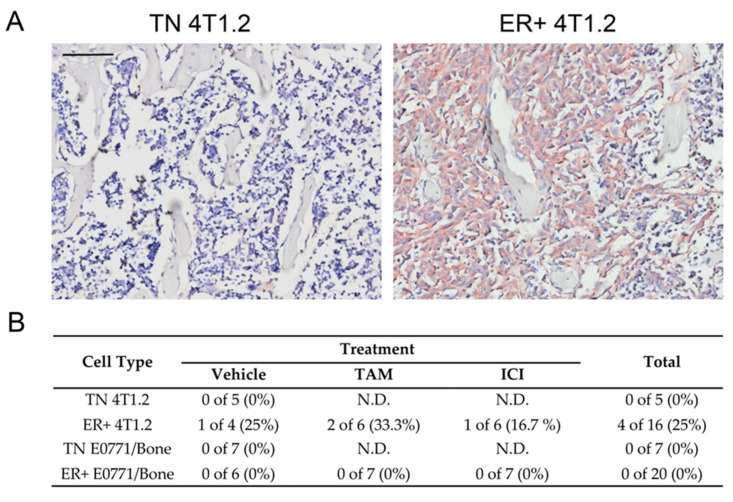
Hind limb metastases were present in mice with ER+ 4T1.2 tumors regardless of antiestrogen treatment. Examples of immunohistochemistry of tibiae from mice (*n* = 48) for pan-cytokeratin. Only the ER-expressing 4T1.2 tumors metastasized to the bone, regardless of antiestrogen treatment. (**A**) Examples of TN 4T1.2 and ER-expressing 4T1.2 tumor pan-cytokeratin immunohistochemistry at the metaphysis are shown. Scale bar represents 150 µm. (**B**) Number of mice with metastatic lesions was quantified as the percentage of positive mice shown in the table for each cell type and treatment group. N.D.; not determined.

**Figure 7 cancers-15-05773-f007:**
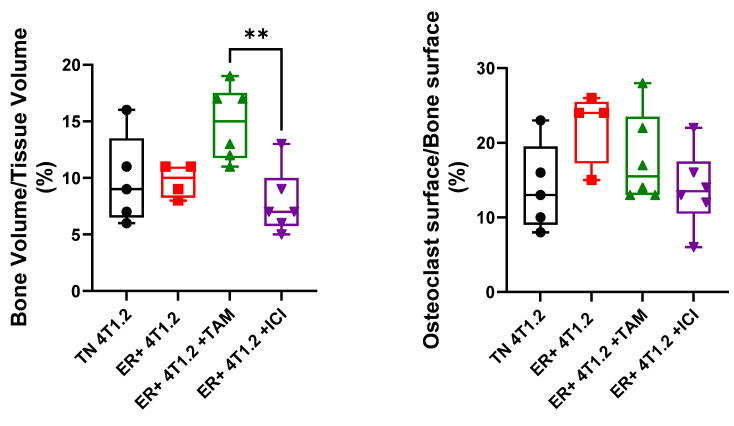
TAM induced alterations in the bone niche. Tibiae were isolated from mice after tumor implantation with ER-expressing or TN 4T1.2 BC cells and treatment with vehicle controls, 5 mg/60-day time-release pellet TAM, or 1 mg/wk ICI. Bones were stained for TRAP+ osteoclasts and bone histomorphometry was performed to calculate bone volume-to-tissue volume fraction or osteoclast surface over bone surface. A significant decrease in bone volume to tissue volume was measured in the mice with ER-expressing 4T1.2 tumors treated with ICI when compared with those treated with TAM. TAM treatment increased the bone volume-to-tissue volume ratio compared with other groups. Both are represented as mean percentage ± SEM (*n* = 4–8). ** represents *p* < 0.01 according to one-way ANOVA.

**Table 1 cancers-15-05773-t001:** There were no differences in metastatic organ tropism between ER-expressing and TN 4T1.2 cells. One H&E-stained, formalin-fixed, paraffin-embedded tissue section of each reported organ was examined per mouse. There was no histological evidence of differences in organ tropism for metastatic lesions between the ER-expressing and TN 4T1.2 tumor-bearing mice (*p* > 0.05, Chi-squared test).

	Lung	Liver	Brain	Heart	Kidney	Spleen	Hind Limb
ER-expressing 4T1.2	15/23 (65%)	1/19 (5%)	0/23 (0%)	4/21 (19%)	0/24 (0%)	2/24 (8%)	2/24 (8%)
TN 4T1.2	13/21 (62%)	4/20 (20%)	0/19 (0%)	3/20 (15%)	3/21 (14%)	4/20 (20%)	5/21 (24%)

## Data Availability

The data can be shared up on request.

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
