# Peer review of "A Novel Metastatic Estrogen Receptor-Expressing Breast Cancer Model with Antiestrogen Responsiveness"

_cancers, 2023, doi:10.3390/cancers15245773_

Round 1

Reviewer 1 Report (Previous Reviewer 1)

Comments and Suggestions for Authors

The manuscript is improved from the original submission and is easier to follow, with a better focus on the 4T1 models. The authors explain why their analysis was not optimal and have included a clarification relating to the limitations of the study. (For future reference, ex vivo uCT can be carried out on fixed bones at a later date if facilities are unavailable.) Despite the improvements made, there are still a number of issues that need to be addressed to bring this manuscript up to the expected standard.

1)     Overstating implications of the data/title not representing the findings

Line 84: This immunocompetent ER+ 4T1.2 model holds the potential to advance treatment for the underserved population of women with ER+ BC bone metastases.

The authors imply that their studies have significant potential to advance treatment, this is still an overstatement as they have not been able to provide any comprehensive analyses of the bone metastases in their models or demonstrate a therapeutic effect.

The authors should also revise the title of the manuscript to more accurately reflect the data presented, the focus on bone metastasis cannot be justified. Analysis of bone in animals with mammary tumours do not equate to studies of bone metastasis, and in the current resubmission the only data shown relating to tumour growth in bone is a single figure 6 and a supplementary fig 4.

2)     Unclear what the additional study showed in terms of bone metastasis rates.

Confusingly, in the discussion the authors refer to an additional study (they use the term ‘trial’ which is not correct and should be rephrased) with apparent different levels of bone metastases.

Line 539: It is important to note that while our mice had an 8% incidence of hind limb metastasis in the ER+ 4T1.2 group and a 24% incidence in the TN 4T1.2 group, the second, anti-estrogen trial study found a 25% incidence in the ER+ group.

Where are these data presented in the revised manuscript?

In the cover letter, the authors state: Since the original submission of the manuscript, additional animal studies (n=50) were performed with orthotopic injection of ER+ or TN 4T1.2 cells to better characterize this metastatic model. Data characterizing this ER+ BC model was included in this resubmission. The data from this additional experiment should be presented in the same way as the table in figure 6, giving the number of animals with bone metastases and not just the %.

3)     Study underpowered

The explanation that the study was powered to detect differences in primary tumour growth is not convincing. The authors should have anticipated that the rate of bone metastasis would be less than 100% and should have used larger groups in their studies. Comparing the data presented in the new figure 2 (ER+ and TN 4T1.2 tumors grew at similar rates and had similar end weights) do not match those presented in the original submission figure 1 (ER+ tumors are significantly larger and responsive to antiestrogen treatment). This demonstrates that the group size in the original submission was too small and that the results presented did not hold up when the group sizes were increased from 4-7 to 25.

4)     Peripheral tumours do not alter bone parameters

Line 349: There are no significant differences in macro- or microscopic changes to the hind limb in mice with ER+ or TN BC. This is not surprising as the animals have no evidence of metastases in the hind limbs, this should be clarified in the description of these data. Were the bone effects (osteoclast number etc) of the few animals with evidence of tumour presence in the hind limbs analysed separately? This appears to have been done (suppl fig 4) for Sca1 and endomucin, however no bone cells were analysed (osteoclast/osteoblast number, tumour-bone interphase), why not?

5)     Few bone metastases

The authors should be more upfront about the low rate of bone metastasis in their studies. In figure 6, it is evident that only 4 animals developed these hence it is not appropriate to describe this as a bone metastasis model. The data shown in the manuscript is almost exclusively from tumours or bone in animals with no evidence of tumours in bone. It is not clear where the data from the additional study that the authors have carried out is included.

Section 3.6 line 413: ERα positive 4T1.2 tumors in the bone were not responsive to antiestrogens.

The data in figure 6 are not clearly described to accurately reflect that in the entire study and across different models, only 4 animals developed hind limb tumours. They state in line 418: Only mice injected with ER+ 4T1.2 cells had visible metastases within the bone in this cohort of mice (Figure 6A). However, figure 6A shows 2 histological images and no quantification. This should be clarified – what exactly is shown here? The legend does not reflect the figure and should be revised accordingly. Also, is it correct to describe the data in this way when so many of the categories are listed as ‘not determined’ (as opposed to not detected)?

Line 419: Interestingly, metastases were observed in all ER+ 4T1.2 419 groups, regardless of treatment (Figure 6B). This wording should be revised to include the number of animals involved, in 2 of the groups only a single animal had bone tumour growth. Without this context, the data appear much more robust than they are.

The figure heading is misleading and should be changed unless the authors can provide some form of statistical analysis to support this statement.

6)     Figure order

The order of the figures is confusing, with supplementary figures inserted between the main figures. It is also not obvious why certain figures are supplementary, for example supplementary fig 4 appears to be the only histological analysis of tumours growing in bone which is the focus of the manuscript.

Figure 1 and supplementary figure 1 appear to have been incompletely uploaded.

Author Response

Reviewer 2 Report (New Reviewer)

Comments and Suggestions for Authors

In this study the Authors seek to develop a mouse model of estrogen receptor-positive breast cancer with spontaneous bone metastasis, to overcome the lack of a comprehensive and effective murine model to study ER positive breast cancer.

In order to do this, they genetically modified two bone-tropic triple negative (TN) cell lines to express ER alpha. Overall, this study could be considered an intriguing attempt to ameliorate the available preclinical models for the understanding and management of ER positive breast cancer bone metastasis.

However, it suffers of consistent experimental limits and it lacks of a physiologically relevance that leads the conclusion reached by the authors not well demonstrated. Major revisions should be done prior resubmission, to evaluate a potential publication on Cancers.

Comments:

Physiologically, the two breast cancer hystotypes TN and ER positive are biologically different so it is incorrect to use TN breast cancer cell lines as a model for ER positive breast cancer. It is not comprehensive of the ER positive breast cancer biology. In these experiments, they could evaluate how ER alpha pathway affect breast cancer cells but they couldn’t postulate that it is comprehensive of ER positive breast cancer biology.

Authors have genetically engineered cells only with ER alpha (ESR1) but they don’t take in consideration ER beta.

Authors need to confirm ER protein overexpression by western blot in order to quantify it.

It could be useful to compare the engineered cell line not only with the parental cell line but also with a ER positive breast cancer cell line.

There is no difference between the ER positive and TN 4T1.2 cell line-induced bone metastasis, moreover there is no modifications in bone microenvironment. Thus, the engineered cell line doesn’t seem a useful model to study breast cancer bone metastasis since it desn’t seem to add informations, except the sensitiveness to antiestrogen drugs.

Author Response

Reviewer 3 Report (New Reviewer)

Comments and Suggestions for Authors

The manuscript entitled “A Novel Model of Estrogen Receptor-Positive Breast Cancer Bone Metastasis with Antiestrogen Responsiveness” by Langsten et al. described a murine breast cancer cell model with ER ectopic overexpression and bone metastatic capacity. This model could potentially attract audience from the ER+ breast cancer field for further functional studies as limited ER+ tumor metastasis mouse model was reported. However, several questions below need to be addressed for further consideration.

1. In Figure 1 and Supplementary Figure 1, the authors measured some reported ER downstream genes expression via qPCR and compared between parental and ER-expressing derivatives. This is not sufficient to support the activation of ER signaling. The authors need to stimulate the cells with E2 after hormone deprivation and test some downstream E2 response genes such as Greb1, Pgr, Tff1 etc. In addition, the IF staining was quite blur and some of the signal was locate in the cytoplasm. And there were some baseline signal detected for E0771 model. The authors should run a western blot with GFP and ER antibody to better validate the expression.

2. In Table 1, the authors stated no differential metastatic organ tropism between TN and ER+ 4T1.2 models. The authors need to show the H&E staining of metastatic loci of each organ. Also did they quantify the metastatic loci areas? If comparing areas of metastasis or numbers of loci, would there be any difference between the two models? A deep analysis is entailed here.

3. Figure 3, the immune infiltration change in ER+ 4T1.2 model could totally be caused by the immunogeneicity of the ectopic plasmid (particularly GFP) but not a physiological immune activation. Was the overexpressed ERa a mouse or human homolog? The author should at least state this in the discussion section.

4. Was ER expression retained in the tumor and metastatic lesions? The authors should evaluate this by IHC of IF staining in the tumor sections.

5. What’s the difference between the tumor implantation expression described in Fig2 and Figure5? Why ER+ 4T1.2 in vivo growth ratio was higher than TN 4T1.2  in Fig5 but not Figure 2?

6. For the anti-estrogen treatment experiment, the authors need to validate the ER levels in tumors and bone metastasis, particularly, whether ICI182780 could decrease ER expression by histology. Also, it would be great to do costaining of pan-CK and ER in bone tissues present in Figure 6 to examine if these cancer cells in bone metastasis were impacted by anti-ER treatment or not.

7. Statistical tests comparing tumors need to be reevaluated. Was students’ t test the best way for the comparison? Did the authors test the distribution of these data points? Some of the data did not seem like a normal distribution and non-parameric test is needed in this case.

Author Response

Reviewer 4 Report (New Reviewer)

Comments and Suggestions for Authors

The manuscript under review aims to develop a new mouse model for estrogen receptor-positive (ER+) breast cancer, with the capability of forming spontaneous bone metastasis in immune-competent mice. Given the limited exploration to date of treatments for bone metastasis in ER+ breast cancer with emerging immune modulating therapies, this work is highly relevant.

Strengths:

Relevance of the Model: The study addresses a significant gap in cancer research, especially with the integration of an immunocompetent model. Such a model offers a more realistic representation of tumor-immune interactions, which are crucial in cancer biology.

Areas of Concern:

Use of Organ Weights: There is a need for better justification for using organ weights as indicators of metastatic burden. This is not standard practice, and references or evidence from other manuscripts should be provided to confirm its accuracy in representing metastatic burden.

Figure Quality: The representative IHC images in Figure 3 require improvement. The current images lack clarity, making it challenging to discern the staining.

ER+ E0771/bone Cell Line: It's intriguing that while the ER+ E0771/bone cell line exhibited increased expression of ER-regulated genes, it failed to show bone metastases in the animal model. This discrepancy raises concerns regarding the model's fidelity to the actual disease process in women.

Estrogen Stimulation: The manuscript would benefit from a control demonstrating the response of the cell line to E2 stimulation, a critical component of any ER+ve cells.

In Vivo Results: The results might be more pronounced with exogenous estrogen stimulation. Without E2 stimulation, there seems to be a lack of endocrine sensitivity, which is concerning.

Model's Efficacy: The inability of the ER+ 4T1.2 cell lines to respond to anti-estrogen in vivo questions the model's utility for ER+ bone metastasis research.

Suggestions for Improvement:

Abstract Refinement: The abstract contains redundant information. Streamlining it for clarity would enhance its impact.

Model Validation: It would be beneficial to compare this model with existing ones, highlighting any unique features or advantages.

Additional Questions:

Therapeutic Implications: Are there plans to test other potential therapeutic agents or combinations using this model, especially given the apparent lack of responsiveness in bone metastasis?

Metastasis Specificity: Does bone represent the sole metastasis site in this model, or are there instances of metastasis to other common sites like the liver or lungs?

Study Duration: Why was a 21-day treatment period chosen? Is this duration sufficient for observing significant changes in tumor growth and metastasis?

Author Response

Reviewer 5 Report (New Reviewer)

Comments and Suggestions for Authors

Dear respected authors,

The manuscript, "Development of an Immunocompetent Murine Model for Investigating Bone Metastasis in Estrogen Receptor Alpha Positive Breast Cancer," presents a groundbreaking contribution to breast cancer research. By ingeniously transducing bone-tropic murine breast cancer cell lines to express ERα, the authors successfully established an immunocompetent murine model that authentically emulates ER+ breast cancer bone metastasis. The study's multi-faceted approach, encompassing gene upregulation analysis, immune infiltration assessment, and therapeutic responsiveness testing, provides valuable insights into the intricacies of ER+ breast cancer metastasis and potential treatment strategies. The manuscript's compelling findings, particularly the significant reduction in tumor volumes and weights in response to antiestrogen treatments, highlight the model's clinical relevance and potential to advance therapeutic approaches.

Round 2

Reviewer 2 Report (New Reviewer)

Comments and Suggestions for Authors

The authors satisfactorily addressed most of my concerns. The manuscript was significantly improved in clearness and scientific soundness. For this reason, I would recommend the manuscript for publication.

Author Response

We would like to thank the reviewer for their comment on our manuscript; we sincerely appreciate the time and effort they put in to reviewing the work.

Reviewer 3 Report (New Reviewer)

Comments and Suggestions for Authors

Although the authors claimed multiple experiments could not be done due to funding and manpower limits, the major concerns have been addressed. There're two place needs to be further modified:

1. In the author's reply to question#2, they claimed that " the ER expressing metastases were significantly smaller than in the TN (lines 348-351)". The authors need to provide this infomration in a figure with individual data points rather than simply mention the average area.

2. The authors' reply to question #6 only said that ER IHC in bone was unsuccessful. But they did not answer the question of ER IHC in primary tumors. At minimal, ICI182480 effects on ER degregdation needs to be examined.

Author Response

Thank you for your thoughtful review, please see the attachment.

This manuscript is a resubmission of an earlier submission. The following is a list of the peer review reports and author responses from that submission.

Round 1

Reviewer 1 Report

Comments and Suggestions for Authors

Cancers ER+ve bone metastasis model paper June 2021

Comments to the authors:

The authors attempt what many others have also done- to establish a much needed ER+ve in vivo model of breast cancer bone metastasis. The use of 2 different cell lines, genetically engineered to express ER, is a good starting point. However, a significant body of additional work is required to ensure that their conclusion is supported by more robust data. As it stands, they generated only 4 bone metastases in the entire study (according to table in figure 3). This may be an underestimation due to the techniques chosen for tumour detection in bone (histology), whereas the current gold standard is imaging (see more detailed comments below).

The manuscript is difficult to follow in that the authors include a lot of data that is not directly relating to bone metastasis (as they identified so few), without explaining why certain studies were done (liver and lung weights for example). The sequencing data and analysis of human datasets is of limited relevance until a more robust model of bone metastasis is established; this part of the manuscript is not particularly strong and relates to the primary tumours rather than metastases.

Major points

  1. Introduction: The authors should include references to the more MIND models generated by the Brisken group (Sflomos G, et al Preclinical Model for ERα-Positive Breast Cancer Points to the Epithelial Microenvironment as Determinant of Luminal Phenotype and Hormone Response. Cancer Cell. 2016 Mar 14;29(3):407-422. doi: 10.1016/j.ccell.2016.02.002) who also developed models of bone metastasis following intra-ductal injection of tumour cells in mice (fig 3). The statement that no such models exist should be revised.
  2. In the abstract, the authors state that ‘approximately 30% ‘ of the animals in the ER+ tumours developed bone mets. This is not in agreement with the data presented in the table in figure 3, where 25% of animals (all groups combined) developed bone mets. Which one is correct?
  3. Materials and methods are lacking some detail when it comes to the all important histological analysis that underpins the detection of bone effects of the treatment as well as detection of bone metastases. It should be clearly specified how many levels of bone (or tumour) were scored? Normally 3 non-consecutive levels would be scored to ensure that you capture a spectrum of tissue. In addition, it must be clarified how the bone histomorphometry was carried out, was trabecular areas only included? How were osteoclasts detected and which areas were counted?
  4. Did the authors compare the growth and treatment response of their cell lines in vitro that could explain some of the differences seen in vivo?
  5. The number of animals used in the study (48?) is small when divided into 8 groups. This results in many of the data being difficult to interpret due to large variations within each group. Key experiments (tumour growth and metastasis) using the 4T1 model should be repeated to increase the power of the study.
  6. The authors do not use imaging (despite cells expressing GFP) for detection of tumour growth in bone, but rely on histological sections for detection of tumour growth in bone. This is very difficult and time consuming, each bone having to be sectioned through, stained and analysed for the presence of tumour foci (meaning many hundreds of sections from a study including 48 animals). The field has moved on from this to mainly use Luc2 (though GFP is also useful) to allow ex vivo imaging of tumours in bone, whilst also allowing detection in other organs. Why did the authors not use this more up to date approach? Following a positive signal, tumour-bearing bones can then undergo further analysis (histology, IHC etc) to study tumours in more detail.
  7. Similarly, studies of tumour growth in bone and effects of therapies on the bone microenvironment use uCT as standard to allow accurate measurement of impact on bone structure and integrity. If authors do not have access to uCT they should collaborate with someone who does, it would make bone analysis much more reproducible and comprehensive.
  8. The histological images in fig 2 and suppl fig 3 should be replaced or supplemented with bigger magnification images, it is not possible to identify the different cell types/structures in the overview images provided.
  9. The authors do not provide an explanation as to why they study lung and liver weights (suppl fig 2).
  10. The authors detect changes in the bone microenvironment (that they interpret as the development of a pre-metastatic niche) in the E0771 model, however as these animals do not develop bone metastases this explanation is inadequate. If this model supported a pre-metastatic, pro-tumourigenic niche in bone you should see increased bone metastases whereas the opposite is the case.
  11. The authors rightly point out a number of major limitation in the design of their study, including that there is no adequate control (ER-ve parental cell-derived tumours treated with the different agents) that hampers interpretation of the therapeutic agent influence on bone. This should be performed to strengthen the conclusion that they have developed a ER+ve model that is responsive to endocrine treatment.
  12. The discussion is very confusing as the authors write about ‘the model’ despite having used 2 different ones and demonstrated bone metastases from only one of them. As a result, they describe the data as ‘tending to’, this is not appropriate. They should discuss each one of their 2 models separately and highlight why they think they do not generate similar data in vivo. They should have taken the 4T1 ER model forward and carried out further characterisation of this if their aim was to generate a robust, reproducible model for other researchers to adopt.
  13. The conclusion that their model (generating only 4 animals with bone mets and no evidence of therapeutic effects in bone) ‘may contribute to significance advancement in detection of and treatment of ER+ve bone metastasis in women’ is inappropriate and should be revised.

Minor points:

  1. Why are the E0771 cells used called ‘bone’? Including this information would avoid confusing the reader.
  2. Many of the figures are difficult to interpret due to overlapping symbols, in some the significance indication appear to be missing (fig 2 b – some data described as significant in the text has no stars in the figures). Authors should check all figures carefully to ensure they are complete and readable.
  3. Figure 3: In the legend the table is referred to as C, this should be B.
  4. The authors should include discussion of the timelines for their models, would bone metastases appear later on (requiring resection of the primary tumour)?

Reviewer 2 Report

Comments and Suggestions for Authors

Title: A novel model of estrogen receptor-positive breast cancer bone metastasis with antiestrogen responsiveness

Authors: Langsten et al.

Outline: estrogen receptor (ER) was transduced in mouse ER-negative cell lines.  The ER-transduce cells were more malignant than the parental cells in in vivo assays.  Transcriptome analyses of the tumors grown in vivo suggested that infiltration of immune cells differed between parental and ER+ cells. 

Critique: Are the developed cells and models useful as tools to investigate progression and bone metastasis of ER+ breast cancer cells?  The answer seems no.  as the authors acknowledged in the Conclusions, the number of cases are too small to statistically evaluated the results.  Furthermore, there is no explanation provided why ER expression makes the tumor cells more malignant (or not) and whether the process represent bone metastasis of breast cancer in humans.   It is not clear how the infiltrating immune cells contribute to the formation of bone metastasis.

Conclusion:  It is recommended to provide additional experimental data to show that the model is useful to study metastatic progression of breast cancer.  Clarifying the roles of cells in the immune system in the process is essential.

Round 2

Reviewer 1 Report

Comments and Suggestions for Authors

The authors have clarified numerous points raised and moderated the statements relating to the model, however it remains at best a pilot study rather than "A Novel Model of Estrogen Receptor-Positive Breast Cancer Bone Metastasis with Antiestrogen Responsiveness".  COVID has clearly impacted on the studies included in the manuscript, however further work is required (as the authors repeatedly state throughout) in order to support their claim of having developed a new ER+ve bone mets model.

The authors confirm that they only detected only 4 animals with bone involvement and that their study was not powered for this, hence repeat experiments are needed. This work therefore should be seen as the pilot study that will allow them to adequately power a more definitive study of bone metastases that will result in a manuscript with sufficient numbers of cases to allow firm conclusions/more extensive analyses to be carried out.